# Crucial Development: Criticality Is Important to Cell-to-Cell Communication and Information Transfer in Living Systems

**DOI:** 10.3390/e23091141

**Published:** 2021-08-31

**Authors:** Ione Hunt von Herbing, Lucio Tonello, Maurizio Benfatto, April Pease, Paolo Grigolini

**Affiliations:** 1Biological Sciences Department, University of North Texas, Denton, TX 76203-5017, USA; AprilPease@my.unt.edu; 2GY Academy Higher Education Institution, E305, The Hub Workspace, Triq San Andrija, SGN1612 San Gwann, Malta; lucio.tonello@gy.edu.mt; 3Center for Nonlinear Science, University of North Texas, Denton, TX 76203-5017, USA; paolo.grigolini@unt.edu; 4Laboratori Nazionali di Frascati, Istituto Nazionale di Fisica Nucleare, Via E. Fermi 40, 00044 Frascati, Italy; maurizio.benfatto@lnf.infn.it

**Keywords:** complexity, non-crucial events, crucial events, FBM, memory, embryos, seeds, development

## Abstract

In the fourth paper of this Special Issue, we bridge the theoretical debate on the role of memory and criticality discussed in the three earlier manuscripts, with a review of key concepts in biology and focus on cell-to-cell communication in organismal development. While all living organisms are dynamic complex networks of organization and disorder, most studies in biology have used energy and biochemical exchange to explain cell differentiation without considering the importance of information (entropy) transfer. While all complex networks are mixtures of patterns of complexity (non-crucial and crucial events), it is the crucial events that determine the efficiency of information transfer, especially during key transitions, such as in embryogenesis. With increasing multicellularity, emergent relationships from cell-to-cell communication create reaction–diffusion exchanges of different concentrations of biochemicals or morphogenetic gradients resulting in differential gene expression. We suggest that in conjunction with morphogenetic gradients, there exist gradients of information transfer creating cybernetic loops of stability and disorder, setting the stage for adaptive capability. We specifically reference results from the second paper in this Special Issue, which correlated biophotons with lentil seed germination to show that phase transitions accompany changes in complexity patterns during development. Criticality, therefore, appears to be an important factor in the transmission, transfer and coding of information for complex adaptive system development.

## 1. Introduction

All living organisms are in a dynamic balance between organization and disorder as they undergo renewal through processes such as development and regeneration. Renewal and maintenance of highly ordered systems is often associated with increased organization and decreased biological complexity (defined as the number of parts composing a system [1]) resulting in system stability, also referred to as morphological (structural) or physiological (functional) homeostasis [2,3,4]. Understanding how the interplay between organization and disorder maintains homeostasis has been mostly obtained from data gathered from biochemical and energy (metabolic products) exchanges [5]. This bottom-up approach of examining molecular mechanisms to explain the evolution of complex behavior and even cognition has been invaluable for systems biology to understand the emergence of biological complexity [4]. Still missing however, is a top-down or synoptic understanding of how information is processed, and how computation is carried out among interactions between cellular networks and their surrounding local (cellular) and global (ecological) environments.

New work from statistical physics [6,7,8] in this Special Issue suggests information transfer offers a new, more synoptic approach to understanding the evolution of complexity in living organisms. As elegantly expressed in the first paper of this Special Issue [7], while examining the system (e.g., cell, tissue, organism or community) is important, a more complete approach to understanding emergence must include the system *plus* the environment. In living organisms, nowhere is this more relevant than during the process of development, i.e., when a single cell undergoes progressive change to form a multicellular organism—be it a human, mouse, bird, frog, fish, fruit-fly or nematode worm. Following the three previous papers in this Special Issue on “*Memory and Criticality*”, we propose in this paper that development of a living organism can be considered an emergent complex adaptive system undergoing cr iticality, and that developing organisms are in a dynamic process of self-organization that generate crucial events, as detected using diffusional entropy analysis (DEA) described in [7,9].

To quote [10] from their textbook, *Developmental Biology*, “…multicellular organisms do not spring forth fully formed. Rather, they arise by a relatively slow process of progressive change that we call development”. For every living embryo, the structural outcome of the developmental program will be a multicellular entity operating as a stable and dynamic society, whose individual members are made up of cells, and whose dual nature can be explained by thermodynamic processes [5,11]. Specifically, thermodynamics is used here as a collective term for the laws governing energy and information transactions that accompany all physical and chemical reactions. The 2nd Law of Thermodynamics is particularly critical to ordered systems, one version of which states that the entropy of the universe increases with any spontaneous change and is expressed as:*S* = *k* log *W*

where *S* is the entropy (derived from the Greek word for ‘in turning’), *k* is the Boltzmann constant in the equation *b* = 1/*kT* where *T* is the Kelvin temperature and *W* is a measure of the number of ways that the molecules of a system can be arranged to achieve the same total energy (the ‘weight’ of an arrangement) and belongs to the domain of statistical physics [12]. In order for living organisms to follow the 2nd Law of Thermodynamics, the word ‘universe’ is key because it emphasizes, as always in thermodynamics, the system together with its environment, as identified by [7]. Therefore, living organisms are devices that generate organized work by drawing on the dissipation of energy in processes such as metabolism and live to increase the disorder in their surroundings [13].

Schrödinger [14] understood that living organisms extract order from the environment and discard disorder (creating negentropy). Prigogine [13] clarified this by noting that the increased disorder of the environment increased the entropy of the system, thus living organisms adhere to the 2nd law of thermodynamics. New ideas from statistical physics propose that the dynamic interchanges between order and randomness are indicators of different patterns of complexity and are dependent on how information is relayed among complex networks [15]. All complex networks may contain a mixture of patterns of complexity, (i.e., non-crucial and crucial events, see below for details) that exhibit different statistical properties, but it is crucial events that determine the efficiency of information exchange [9,15].

Crucial events are generated by criticality, namely by the processes of phase transition from disorder to correlated disorder, affecting key organismal network functions. There is, as suggested by [15], “a subtle connection between informational exchange within and between networks and the complexity (non-simplicity) of those networks”. West and Grigolini [15] replaced the term complexity with non-simplicity and explain their reasoning by stating that in physics it is easier to understand how phenomena function by the properties or characteristics that are missing, rather than those that are present. Yet, understanding how organisms respond to change (adapt) is heavily dependent on how organisms develop or evolve new emergent behaviors or functions that increase the probability of survival and ultimately reproduction. In short, while physics focuses on what is missing, biology prefers to identify what new processes or responses have been added or modified through natural selection and this is called adaptation.

Adaptation has in the past, always involved processes that generate mass and energy in a living organism. These processes are integrated through complex metabolic networks made up of cellular constituents and reactions. These cellular constituents and reactions respond to external changes, which are then acted upon by selective pressures that occur at and above the genome level. The developmental biologist C.H. Waddington [16,17] first coined the term ‘epigenetics’ to describe this entire process. Consequently, the identification and characterization of system-level features and approaches to biological organization have become a key part of post-genomic biology, dominating for nearly a century because of their bottom-up approach in explaining how living organisms function. We have greatly benefitted from these reductionist approaches since the discovery of DNA [18]. However, reassembly from the constituent parts into a functional organism has been unsuccessful and requires a different, synoptic (top-down) approach offered by complexity science, nonlinear statistical physics and new approaches of criticality detailed in the papers of this Special Issue [6,7,8], including the present paper.

In physics, a system is considered complex if it meets the anti-reductionist criterion of “the whole being greater than its parts”, but as Melanie Mitchell states in *Complexity*, there is really no quantitative definition of complexity [19], and certainly not one upon which physicists, computer-, or biological scientists agree. There does seem to be agreement that living systems exhibit complex, changing behavior at the whole organism level (macrodynamics). This complex organismal behavior emerges from the collective actions and interactions at lower levels of organization, such as among cells (microdynamics). These actions can be understood using dynamical systems theory, which can be described simply as dynamic, i.e., systems that change over time (e.g., heart beats, firing of neurons in the brain, economic markets, or global climate). Developing systems also change in space as well as time and exhibit dynamic patterns of complexity, which adapt in response to learning or external forces. For example, increasing temperature will alter rates of development and growth in fish embryos, expressed as changes in morphological [20,21] and physiological [22] complexity. Complex adaptive systems, therefore, are complex systems exhibiting nontrivial emergent and self-organizing behaviors [19], many of which result from cell-to-cell communication at many developmental levels.

Cell-to-cell communication and its influence on development is generally understood to result from concentration dependent gradients of biochemical molecules called morphogenetic gradients, (see Section 3) driving energy exchange and leading to differential gene expression and cellular differentiation. However, new ideas derived from nonlinear statistical physics suggest that in addition to biochemical gradients there are also gradients of information that may be responsible for initiating spontaneous self-organization, as well as disorder [4,15,23]. New ideas from Michael Levin and colleagues [3,4] also suggest that endogenous bioelectric pathways may play a part in implementing high-level pattern homeostatic states and control loops in developing systems. We suggest here that these bioelectric pathways may also generate crucial events and need to be further investigated.

Also possible, is that a breakdown in cellular organization may be a loss in cell-to-cell communication or information transfer that can be identified by changes in patterns of complexity. This loss in complexity can signal a change of state, or “a loss of complexity”, where a loss of complexity could mean an organizational collapse, perhaps heralding a shift into a new order or state [7]. For example, the authors of [24] used modified DEA (MDEA), to investigate spectra of the heartbeats of autonomic neuropathy patients that exhibited 1/f noise and showed an increase of the scaling parameter from μ = 2 (healthy conditions) to μ = 3 (on the border of ordinary statistical physics) as the disease progressed. Such shifts in criticality may also accompany bioelectric pathways, which when altered, reveal changes in pattern memory in planaria (worm) and deer antler regeneration [25].

In this review, we not only consider the different patterns of gradient exchange (i.e., biochemical or morphogenetic gradients, energetic and informational (entropy)), and their importance in cell-to-cell communication during development, but also report on new ways to detect crucial events. As detecting and quantifying changes in patterns of complexity is difficult, we specifically refer to work in [6] that uses the technique of recording the emission of ultraweak photonic emissions (UPEs) or biophotons. Biophotons can be regarded as an index of thermodynamic activity [5] and changes in emission rates over time (usually hours) results in changes in parameters of scaling [6]. When biophoton emission rates are used in conjunction with an analytical technique called diffusion entropy analysis (DEA) [7,26,27] biophotons have the potential to document dynamic changes of complexity in a developing organism or complex adaptive system.

## 2. Exploring Criticality in Developing Organisms

It is our assertion that the stable yet dynamic society of a developing organism (i.e., plant seedling or animal embryo) results from, or may be initiated by, the dynamic interplay between two patterns of information exchange or complexity. These two patterns are, (1) self-organization or non-crucial events (Fractional Brownian Motion, FBM) [28] and (2) crucial events (defined as events that determine the efficiency of information exchange [7,15,29]. The type of response depends on the interplay of a network of intracellular (within a single cell) and/or intercellular (between cells) communication and their emergent relationships with their surroundings.

Much of what we know about patterns of complexity has been learned from long time series generated from electroencephalographs (EEGs) of firing rates in the human brain [30,31], heart rate variability [24], and swarming birds [8,32,33] among others. In all these examples there are elements of nonlinear interactions, which result in measurable patterns and modes known as phase transitions where there is a dynamic balance between order and randomness, and crucial events. Crucial events in the above situation are also referred to as renewal events, which are events that reset the memory of the system, erasing the memory of, and independent of, earlier events [15,29,31].

Many complex processes can be characterized by crucial or renewal events and all are independent of the underlying microdynamic emergent behaviors that can be localized in time. In [6] the authors clearly laid out that probability distribution densities (pdd) of the time distance, between two consecutive renewal events, which is given by the waiting time distribution $\psi \left(\tau \right)$ and written as an inverse power law,
$$\psi \left(\tau \right)\propto \frac{1}{\tau^{\mu^{\prime }}}$$
where *μ* is the complexity index which can range from 1 to ∞, with complexity occurring when 1 < *μ* < 3. The breakdown in the ergodicity of a complex system for *μ* < 2 is a direct result of the occurrence of crucial events. However, if an event occurs at a specific time, after which subsequent events are produced, it is time dependent and the $\psi \left(\tau \right)$ has a hyperbolic form, see [15] for a more detailed discussion. As is stated in [6] *μ* < 3 suggests a departure from the condition of ordinary statistical physics to nonlinear statistical physics. Further, crucial events can arise spontaneously in complex systems, in keeping with the theory of self-organized temporal criticality (SOTC) [7,9,34,35,36] which posits that a system of interacting units may spontaneously generate temporal complexity, that is self-organized criticality (SOC) characterized by crucial events in time [6] in which *μ* is not limited to merely the non-ergodic regime of *μ* < 2 but extends to the whole complexity range of 1 < *μ* < 3.

In a developing organism, formation of an orderly multicellular network from relatively homogenous material in a single cell is the result of transactions among nonlinear, self-similar and self-organized components. Those transactions are generally known as cell-to-cell communication and operate based on an inducer (e.g., a cell that produces a signal) and a responder (a cell that responds to the signal by changing some behavior). Successful communication takes place when competence occurs, i.e., when a signal results in a response. We will now consider the processes that are understood to regulate cell-to-cell communication and highlight some of the gaps. These gaps in knowledge about cell-to-cell communication may be where information transfer and crucial events could play a part in directing development of coordinated causal multicellularity.

## 3. Cell-to-Cell Communication, Complexity and Self-Organization

While development is a process that occurs in all organisms, it is concerned with more than just cellular differentiation because different cell types of an organism do not exist in random arrangements. In the mid-twentieth century two biologists, Townes and Holtfreter [37] predicted that embryonic cells could have differences in the components of their cell membranes which allowed them to form organs. Now we understand that formation of organs is a result of cell-to-cell communication achieved through biochemical molecules that are secreted or located in the cell’s membrane. These ‘informational’ molecules can bind to receptors on neighboring cells and stimulate a signaling cascade of intracellular reactions, which results in changes in gene expression, enzymatic activity, and cytoskeletal organization, affecting cell shape and cell behavior. However, it should be mentioned that even at the biochemical level, cell-to-cell communication is much more complicated than suggested above. In addition to signaling cascades with the cell, there are also important intercellular secretory products that trigger cellular responses. These responses are typically ligand–receptor based and range from long distance (endocrine) hormones that travel through the blood stream to short distance (paracrine factors e.g., FGFs) that diffuse between cells across the extracellular matrix (ECM) to target receptors on the cell membrane. Also important to cell-to-cell communication are adhesion molecules that mediate the interactions between cells and the ECM, are critical for maintaining cell structure and function, and are key to organization of cells into tissues and organs.

Following fertilization, the process of cleavage transforms a single cell into a multicellular organism containing hundreds of cells (e.g., the nematode worm *C. elegans* contains 946 cells) or trillions of cells in an adult human being. These different cell types then work together to form a biologically complex, coherent, functional organism that can respond to change and exhibit a degree of resilience [38]. Physical (free diffusion, osmosis, viscosity, elasticity, and viscoelasticity) and cellular processes (mentioned above) act on single cells and take part in acts of aggregation and adhesion to form multicellular systems (tissues) and in the process they ‘re-enact’ the development of cell-to-cell communication systems that emerged 1.5 billion years ago [38]. This means biochemical and energetic processes involved in cell-to-cell communication that first evolved 1.5 billion years ago are conserved, remaining essentially unchanged over evolutionary time. This then begs the question that if there are, as we suggest in this paper, informational exchange mechanisms also critical for development, might they too also have been conserved? During later stages of development, when organization of tissues and organs is taking place, the embryo stays in a relatively stationary state of reduced information (entropy) exchange that is maintained for extended periods of time, making self-organizing or autopoesis of living matter possible [39]. Autopoesis is understood to be any increase in the order within the system (i.e., production of negentropy) and is possible only if high internal biological organization through cell-to-cell communication exists [40].

Kauffman, in his landmark paper [41] stated that a fundamental task of biology is to account for the origin and nature of metabolic stability in living systems in terms of the mechanisms that control biosynthesis. Kauffman stated that biosynthesis includes the renewal or new production of cells resulting from a state of disorder through mitosis (which includes both DNA replication and cell division). He goes on to contrast order and chaos as interpreted in physics and then in biology. In physics when considering the thermodynamics of gases, the mathematical laws of statistics bridge the gap between the randomness of colliding molecules and the simplicity of the gas laws. Whereas in biology a gene can specify a protein and that protein can, in turn, control the expression and or repress another gene [42]. In living organisms, mathematical laws also engage large networks (referred to as gene regulatory networks (GRNs)) of interacting genes to bring biosynthetic or biological self-organization from disorder.

Initially a gene was considered to be a binary device, an on-off switch, but each gene is also part of a large network of genes and can act as a repressor or activator (de-repressor) of other genes in a cybernetic loop [41,42]. As developing embryos are thermodynamically open systems “gathering’ materials and energy from the surrounding environment, their gene-regulatory networks can also be autocatalytic, exemplified by a positive or “forward feedback” autocatalytic cycle. Autocatalysis works similarly in biological, physical, and chemical systems and Lotka [43] developed a series of nonlinear equations to simulate autocatalysis. These were then adopted to describe the cycling of relationships (e.g., predator–prey relationships), linking biology, physics, and chemistry. In chemistry such cycling behavior can be seen in the Belousov–Zhabotinsky reactions, where gradients of chemicals induce the responses, and could be regarded as a type of chemical communication [44].

The biologist, mathematician, and classics scholar, D’Arcy Thompson in 1917 also searched for physical principles that would explain biological structure using measurements and scaling relationships. While his book On Growth and Form [45] is a remarkable treatise on the diversity of form in the animal kingdom, it did not fulfill his intention of finding general laws underlying form and structure. Almost a century later a paper by Tusscher [46] described common design principles among segmentation patterns of somites (blocks of tissue that give rise to muscles) and plant root growth dynamics. Both plants and animals use multi-component signaling systems combining systemic and local signals to fine tune and coordinate organ growth across the body. These signaling systems set up concentration gradients of biochemicals and GRNs, whose concentrations initiate differential gene expression. These biochemical gradients are referred to as ‘morphogenetic gradients’ because of their implied function for morphological or structural change at the macroscale from the microdynamic (gene level and epigenetic—above the gene level) (Figure 1).

The concepts of the morphogenetic gradient and morphogenetic field have been around since the 1920s, pioneered by Hans Spemann [47], and revolutionizing developmental biology. Spemann demonstrated that one part of a frog’s embryo could, when grafted into a second host embryo, induce a complete second vertebral axis. These data and other observations led Spemann to introduce the field concept into embryology. He referred to it as a “field of organization”, now known as a morphogenetic field. A morphogenetic field is created from a group of cells whose position and fate are specified with respect to the same set of boundaries [47,48]. A specific field of these cells will produce a specific organ (eye, forelimb, hindlimb, etc.) and its identity will remain intact even when transplanted to a different part of the embryo. Known in biology as modularity or “biological coherence” [10], it is different from quantum coherence in physics, (see [7,8,9] for discussions of quantum coherence). What is not understood is what initiates the formation of morphogenetic gradients. As all living systems represent a balance between stable states near-equilibrium, which are maintained by a gradient of flows of energy and/or matter from the environment [48] and non-linear steady states. Non-linear states can be chaotic and capable of generating ordered structures. In [39], Rossi et al. explain this as follows: “non-linear, far-from-equilibrium systems reveal dynamic order, are unpredictable (non-deterministic), produce higher states of entropy as well as cybernetic loops and feedback mechanisms and by fluctuation processes can pass from one steady state to another”. Waddington [16,17] first illustrated this concept as a stability landscape in which there are basins of attraction (blue ball in Figure 2) under different conditions, such that a small disturbance such as a stressor may induce a shift into a more stable basin of attraction (Figure 2).

Waddington used the stability landscape to describe the cellular development where the cell was represented as the ball at the top of the landscape (Figure 2). The paths available to the ball (or features of the landscape) are determined by the genotype, interactions among cells, tissues, organs, and the environment [8,49], forming what is referred to as the epigenotype (epi- or above-the-genotype). Epigenetics can include all effects and modifications that are dependent on genetic factors, such as DNA sequences, but may increase or decrease phenotypic (observable traits) variation expressed by target sequences in response to environmental cues, or emergent interactions during development [50,51]. The epigenotype can integrate information from external sources and influence development to produce a cohesive (coherent or stable) organism that will adapt to its environment by responding to change.

At the molecular level, however, transitions among states are influenced by molecular dynamics, generating maximum entropy production, which in turn establish stable states that will ultimately dissipate entropy to the environment through dynamic space-time-coherent processes through events in the cell cycle. A system, therefore, tends to reach equilibrium by reducing energy, order, and information content. The decrease of ∆*G* (free energy) in a system may reflect either an energy decrease, or an entropy increase (increase of disorder), as expressed as a thermodynamic equation:∆*G* = ∆*H* − *T ∆S*
where ∆*H* is the change in enthalpy or heat, *T* is the temperature in absolute Kelvin and ∆*S* is the change in entropy. Order and organization are dependent upon forces that direct cell movement based on the thermodynamic model of cell interactions to arrive at a state of the least free energy that is the most thermodynamically stable (homeostatic) state (e.g., the cell adhesion theory see refs, in [10]). This state, while stable, is still dynamic, undergoing exchanges in energy and information with the surrounding universe or environment. It is yet unknown what forces set up or generate these dynamic stability landscapes (Figure 2) or morphogenetic gradients (Figure 1). We suggest that information transfer gradients related to bioelectric fields [3,4], may play a part. But here we must turn to empirical data collected over the last century and more recently by Benfatto et al. [6] using ultraweak photonic emissions (UPEs) or biophotons. Biophotons have largely been ignored as a potential diagnostic tool to detect information transfer (crucial events) or to discern changes in patterns of complexity.

## 4. Measuring Complexity Using Ultraweak Photonic Emission (Upes)/Biophotons

Fröhlich [52] argued that organisms are made up of strongly dipolar molecules packed densely together in a ‘solid state’ in which electric and elastic forces will constantly interact. Macromolecules such as proteins, nucleic acids, and cellular membranes vibrate at characteristic frequencies [53] that result from the coupling of electrical displacements to mechanical deformations. Collective modes (coherent excitations) of electromechanical oscillations (phonons, or sound waves in a solid medium) and electromagnetic radiations (biophotons) that result that can extend over macromolecular distances within an organism [39]. Despite knowledge of these alternative ‘communication’ processes, few other ways have been considered important for cell-to-cell communication pathways in organisms until recently when biophoton research proved useful in human health monitoring [54] as well as identification and treatment of disease, especially cancer [55].

As sometimes happens in science, increasing interest especially for the treatment of human disease re-discovers work completed much earlier. This is the case with biophotons. In the 1920s Alexander Gurwitsch had already turned developmental biology on its head looking for top-down control of embryogenesis using what he referred to as “mitogenetic radiation” (biophotons), and then his work faded into the background. Almost a hundred years after Gurwitsch [56] first reported his “mitogenetic radiation” and the first case of non-chemical distant interactions, in a recent comprehensive review, Volodyaev and Beloussov [57] brought together data collected over the intervening century on biophotons, also known as ultraweak photonic emissions (UPEs). They reached two conclusions, (1) that the UV fraction of UPEs are regarded as real and, (2) the biological effects are difficult to reproduce reliably. While the review [57] does an admirable job of collecting together much of the empirical data on biophotons, some which are supportive of the mitogenetic effect (MGE), there appears to be some ambiguity about the usefulness of biophotons as a non-invasive method for research in biology and medicine and for their importance as a diagnostic tool for health assessment.

Instead of focusing on biophotons as a diagnostic tool for human disease we continue the good work of [57] and highlight new ways in which biophotons can help reveal the very nature of the beginnings of life. Here we highlight the work of [6] and the importance of using biophotons in conjunction with DEA as tools to detect and monitor changes in patterns of complexity over time, especially in a developing organism (a lentil seed). In 1923, Gurwitsch knew that in embryonic development, the number of cells increased as they divided. He wondered what triggered the cells to reproduce in the first place and, through observation, came to the conclusion that there must be coordinated external (environmental) and internal (cellular) events [56]. If this were true then the cell membrane must contain structures, what he called “receptors” that would perceive, and more importantly, convey a signal for a response. Today we describe the conveyance of a signal, as being part of a signal transduction system and these systems, like those of exchange mechanisms, have been conserved over evolutionary time for cellular communication [10]. Yet, the external stimulus that would initiate cell division was unknown until Gurwitsch [56] referred to one form of this information transfer as mitogenic radiation. Most of this understanding resulted from his famous onion tip experiment in which cells in the apical meristem or root tip of the onion seemed to increase the rate of cell division of a neighboring root tip. Gurwitsch suggested that such mitogenetic radiation acted as a non-mechanical deterministic principle in mitosis setting up a field of biophotons. This field resulted in changes in mitotic activity including molecular interactions within the cell cycle and, in fact, Gurwitsch had identified a type of matter and field dualism. In [5] Van Wijk later referred to this interplay between fields and matter, as a substance/field dualism, which in a biological system is based on its associated radiant (electromagnetic) structure, a stationary field obtained by superimposition of all fields associated with sub-systems, where interferences of oscillations are one process of conserving information (in the case of biophotons it is an optical code). The generation of codes associated with information in relation to interference and resonance are pivots for creating complexity, order and organization.

In [6] Benfatto et al., have for the first time correlated biophoton emission rates with changes in developmental organization (lentil germination) using photomultipliers in an experimental technique to detect low intensity photons with a small signal/noise ratio. Benfatto [6] first measured the dark count without seeds in the chamber, obtaining a baseline emission rate and then measured the biophoton emission counts with the seeds in the chamber continuously over a period of 72 h (Figure 3). Using the technique of DEA, dark count could be described by ordinary scaling, suggesting that no patterns of complexity were present in the absence of the seeds in the chamber (Figure 3). Without seeds in the chamber and in the dark a monotonic decrease in photon emission occurred a few hours after closing the chamber to detector noise, due to a reduction from the residual luminescence from exposure to light of the chamber materials (cotton bed and Petri dish) during experimental set-up. In the presence of seeds (*n* = 75), but still in the dark, the photomultipliers picked up a wide range (or bunch-type structure) of biophoton emissions per measurement. The bunch type nature resulted from an experimental artifact from the filter wheel (see [6] for experimental details) which stayed in each position (the wheel contains seven different filters) for one minute producing a bunch of emissions per filter every 443 s. It is also possible that the scattering per filter reading was also due to the range of different rates of development within the seeds during germination as well as variation in the timing of radicle (tap root tip) emergence (Figure 3).

Benfatto et al. [6] divided the total acquisition time during the experiment of 72 h into six regions for DEA analysis. In the first 30 h (regions R1–3, Table 1), DEA without using a technique called stripes (see [6] for equations) and showed that scaling parameters of the experiment had a mean ± SD (σ, SD = standard deviation) scaling index, η, of 0.77 ± 0.03 with a variance (σ^2^) of 0.001, which corresponded to a mean μ = 2.30 ± 0.05 with a σ^2^ of 0.002. While in the second set of three regions (regions #4–6, Table 1), there was a mean ± SD scaling index of η of 0.72 ± 0.02 with a σ^2^ of 0.0005, which corresponded to a mean μ = 2.39 ± 0.04 with a σ^2^ of 0.002. Thus, DEA without stripes could not distinguish between crucial events and FBM (Figure 3). In contrast, when applying the use of stripes to DEA, the results showed a clear and significant time dependence in the first three temporal regions where the mean scaling parameter of η ± SD is 0.56 ± 0.04, and a variance of 0.001, which corresponds to a mean μ ± SD, 2.79 ± 0.11 with a σ^2^ of 0.012 (Table 1). While in the second three regions (regions #4–6, Table 1), there was a mean ± SD of the scaling index of η of 0.503 ± 0.01 with a σ^2^ of 0.00006, which corresponded to a mean μ = 2.99 ± 0.03 with a σ^2^ of 0.0007. For DEA with stripes, results indicated that crucial events exist in the first three temporal regions (#1–3) and non-crucial events dominate in the last three temporal regions (#4–6). Although, as Benfatto et al. [6] explained that there was the possibility of no crucial events and that anomalous scaling (with the significant difference from ordinary scaling η = 0.5) is due to FBM. It is clear however, that no temporal complexity exists during the dark count part of the experiment compared to when the seeds were in the experimental chamber.

To examine the relationship of biological development of the lentil seeds with the recorded biophoton emission patterns and their implication for the existence of crucial events, we will propose that in the first three regions (#1–3) the lentil seeds are undergoing internal preparations with the cell cycle for germination. During this time, each seed uses the energy store or cotyledon within the seed to prepare for germination by replicating DNA as part of the cell cycle (S phase) and these may result in state changes, which generate crucial events (in the first regions). After this, which cell-division (M-phase) occurs with the emergence of the radicle (tap root) (germination) (in the last three regions), when crucial events were not detected.

In short, this landmark work from Benfatto et al. [6], is the first empirical data gathered to show that the developmental process of germination of lentil seeds is a process that has phase transitions accompanied by changes in patterns of complexity (crucial to non-crucial events). Past work using biophotons to detect changes in developmental processes has been shown in yeast growth, with biophoton intensity being higher during DNA replication (S-phase) than during cell division (M-phase) [58] and in wheat seeds [59]. However, in both cases it was not possible to distinguish if different patterns of complexity existed because DEA was not applied to their data. Benfatto et al. [6] concluded that using DEA as the statistical analysis in conjunction with changes in biophoton emissions suggests that criticality appears to be an important factor in the transmission, transfer and coding of information important to multi-component processes in living organisms, such as germination in plant seeds. These results offer the possibility of using biophoton and DEA in experiments to investigate the existence of variation in complexity patterns in a variety of different developing organisms. Further, these data provide evidence for the importance of the exchange of information (entropy) transfer for cell-to-cell communication during organismal development.

## 5. Conclusions and Future Directions

The main contribution of this fourth paper to the Special Issue is the creation of a bridge between the theoretical debate on the role of memory and criticality, widely discussed in the three earlier manuscripts [6,7,8] with key concepts in biology, and with a special focus on cell-to-cell communication in organismal development. To help readers establish this important connection with biology, which is an important goal of this Special Issue, we note here the pioneering work of Turing [60] and Prigogine [13,61], which led to the discovery of pattern formation and self-organization processes. As suggested in the present paper, these pattern formation processes serve as important evidence that biological order in organismal development is not incompatible with the second principle of thermodynamics. Rather, the generalized thermodynamics of Turing and Prigogine led us to investigate phase-transition processes as sources of temporal complexity and to shed light on the memory-versus-criticality issue itself. In fact, the current literature on the works of Turing and Prigogine emphasize oscillations and instabilities at the same time, thereby suggesting that a combination of criticality-induced crucial events and coherence is possible [7,8]. The adoption of fractional derivative formalism advocated by West and Grigolini [15] does not involve only crucial events but can also be used to generate damped oscillations mimicking the prey–predator behavior of Turing dissipative structures [62].

This fourth paper is also connected to the important issue of cognition referred to in the third paper of this Special Issue [8]. The third paper examines the emergence of cognition, whether is it of quantum mechanical nature or is determined by criticality-induced crucial events. In addition, the present paper emphasizes the possible importance of biophotons [6] for cell-to-cell communication that is expected to be a source of cognition [3,4,63]. In fact, Tonello and his co-workers in 2015 used the arguments of Craddock et al. [64] to advocate a possible way to the emergence of cognition. The work of [63] is based on quantum mechanics, but the readers should also consider the important role of criticality and crucial events to generate cognition and bounded rationality [8]. Another important complexity property discussed by this fourth paper is autopoiesis [39,65]. The biological self-organization termed autopoiesis by Maturana and Varela [66] has been extended to social self-organization [67], thereby affording further support to the important role of criticality. In fact, the authors of [68] have adopted criticality to explain the emergence of social cognition, a property that seems to be related to social autopoiesis.

Finally, the present paper explored the dynamic nature of organismal development, in which there are clearly dynamic stable states or equilibria resulting from changing patterns of complexity. As a result, our understanding of homeostasis, as introduced at the beginning, must change to a less static view, choosing terms such as dynamic homeostasis or allostasis (stability through change [2,69], already in use in the physiology literature, to better reflect the nature of cellular communication at all levels, biochemical, energetic, and informational. Here biophotons also play a part, as they are undoubtedly linked to bioelectrical signaling, as bioelectricity is one layer of the multicomponent morphogenetic field or possibly also a bioelectric code [3]. The word code here implying messages, senders (inducers), and receivers (responders), where bioelectric signals interact with chemical gradients and physical forces producing numerous bi-directional and information transfer loops (e.g., cybernetics and gene regulatory networks (GRNs)).

Intriguingly, the authors of [3,4] showed that by altering bioelectric fields around developing organisms, they altered the morphogenetic gradients, changing developmental trajectories. To test the relationship between bioelectric fields, information transfer, and criticality, might involve modifying the bioelectrical fields at the same time as monitoring mitotic activity, ion-channel activity (linked to bioelectric fields), and biophoton intensity in a developing organism. Either a developing lentil seed or an embryonic zebrafish would be good candidates for this exciting work, as we need to further refine the distinction between the dynamic state of organization and disorder and crucial events, and their importance in developing systems. While discussion of this work is beyond the scope of present paper, finding links between bioelectrical field manipulations and biophoton emissions may further reveal the origin of biophotons, which is still largely unknown (see [6] for further discussion of this topic). In summary, this fourth paper has the important merit of helping the researchers to organize further scientific investigations by using the process of development in organisms to settle the problems discussed in the earlier three papers of the Special Issue of Entropy on memory and criticality.

## Figures and Tables

**Figure 1 entropy-23-01141-f001:**
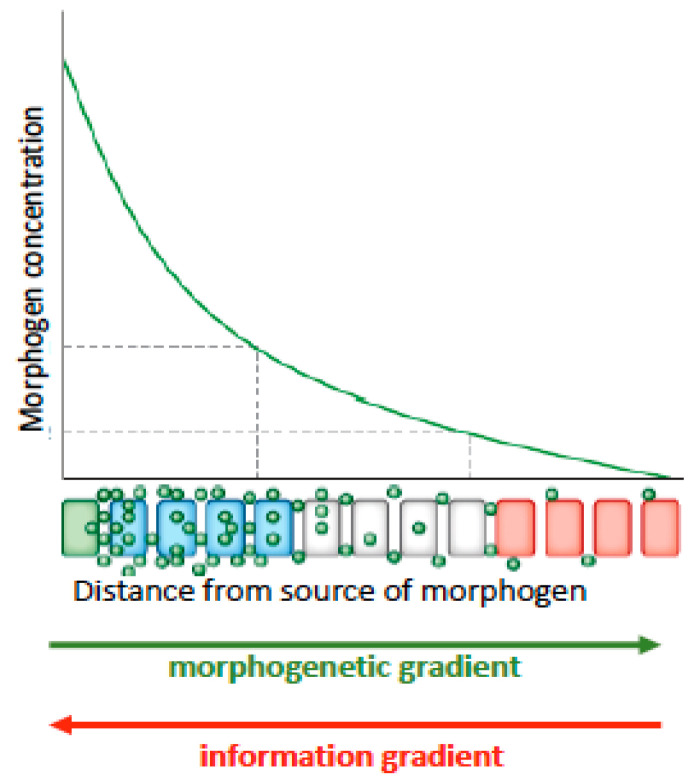
A biochemical morphogen is secreted from a cell (green) and forms a concentration gradient that decreases with distance from the source (green arrow). Dashed lines represent potential threshold points that result in gene expression leading to cell differentiation. Red arrow represents a possible information (entropy) gradient which flows in the opposite direction to the morphogenetic gradient. Figure is modified from [42].

**Figure 2 entropy-23-01141-f002:**
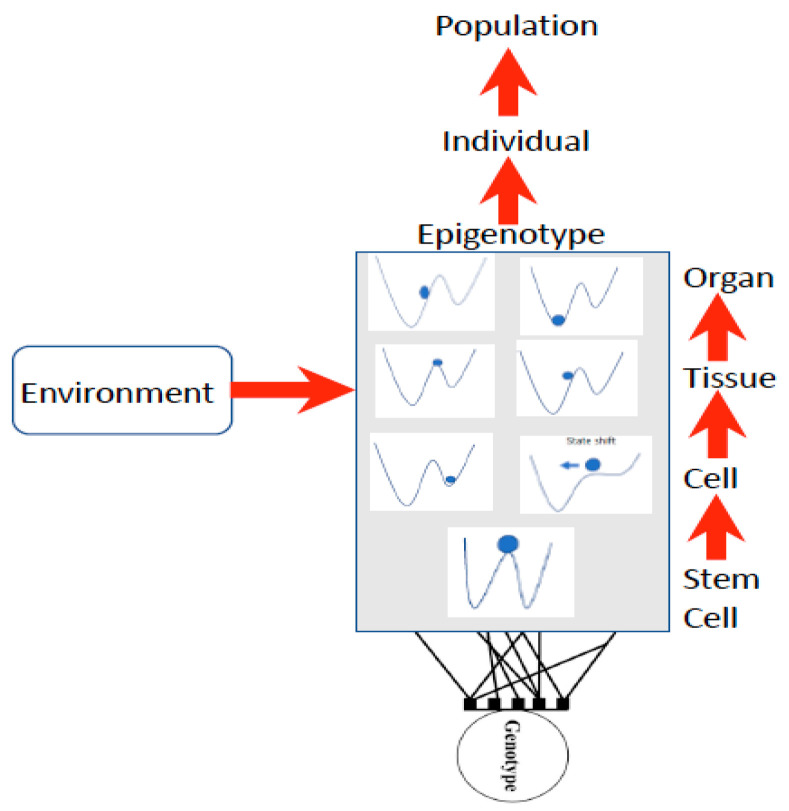
Waddington’s epigenetic landscape shown at each level of developmental organization from stem cell to organ as a complex network of epigenetic interactions defined by the genotype and the environment and influencing individual and population dynamics. The blue circle is the dynamic state under selection, being attracted to valleys or areas of highest stability (low entropy). Modified from [1].

**Figure 3 entropy-23-01141-f003:**
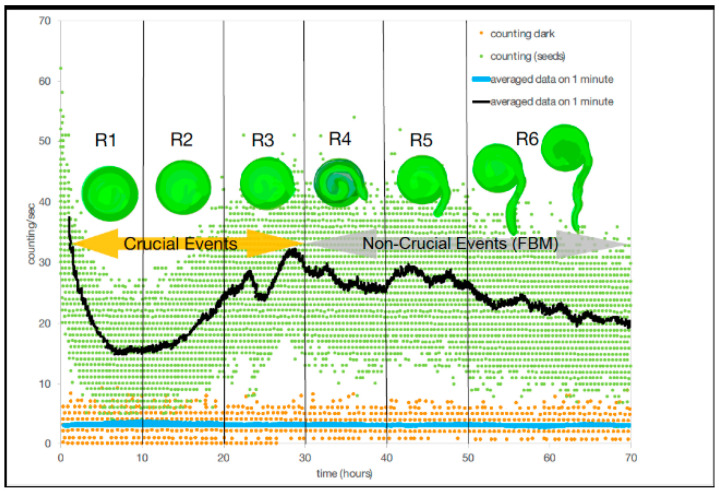
Graph shows the signal or counts of biophotons vs time (h) generated by lentil seeds (n = 75) within a dark chamber (see [6] for experimental set-up). Raw data are shown, seeds (green) and dark counts (red) and the time averaged raw counts over one minute are shown. Vertical lines represent six regions (R1–R6) used in the DEA analysis. Middle panel shows artist’s rendering of lentil seeds before emergence of the radicle (tap root) (i.e., regions R1–3) which is the region of crucial events (orange arrow) and emergence and growth of radicle (i.e., regions R4–6) and region of non-crucial events or FBM (grey arrow). Figure is modified from [6].

**Table 1 entropy-23-01141-t001:** The scaling factors obtained using DEA with and without stripes for six different regions of the total 72 h time series including with seeds and dark count regions. Regions 1–3 represent regions of crucial events, Regions 4–6 represent regions of FBM. Table modified from [6].

	Without Stripes		With Stripes	
	* η *	*µ*	* η *	*µ*
Dark Counts	0.575	2.739	0.508	2.968
1	0.777	2.293	0.596	2.677
2	0.796	2.254	0.558	2.792
3	0.736	2.358	0.526	2.901
4	0.737	2.355	0.496	3.016
5	0.694	2.440	0.509	2.964
6	0.725	2.377	0.504	2.984

## Data Availability

Not Applicable.

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
