# Peer review of "Crucial Development: Criticality Is Important to Cell-to-Cell Communication and Information Transfer in Living Systems"

_entropy, 2021, doi:10.3390/e23091141_

Round 1

Reviewer 1 Report

In this review a keystone concept in biology like cell-to-cell communication in organismal development is addressed considering information transfer. During key transitions like embryogenesis, cell-to-cell communication leads to morphogenetic gradients that produce differential gene expression. The authors propose that in these transitions also appear gradients of information transfer that give rise to stability and disorder loops, associated with adaptive capability. The proposal reflects the nature of cellular communication at all levels, biochemical, energetic and informational and even suggest the use of terms such as dynamic homeostasis or allostasis instead of the more static homeostasis.

In summary, a clear and admirable review about the dynamic state of organization and disorder and crucial events, and their importance in developing systems.

I strongly recommend acceptance.

Author Response

Comments for Reviewer 1.

Thank you so much for your very favorable review. I know all the authors of this paper are grateful for your time as well as your insights.

In accordance with your suggestions, we have thoroughly checked the manuscript and removed any typos or other errors that you mentioned existed in the original manuscript and, thoroughly proofed the manuscript.

Thank you again for your strong recommendation for acceptance in Entropy as part of the special issue “Memory and Criticality”.

Reviewer 2 Report

this review is written about the crucial roles of cell-to-cell communications x for transmission of information in adaptive  systems of embryonic development.  This is very interesting and important for an understanding of mechanism of development by cellular interactions. It was better for adding some sentences, such as 1) important secretion factors, 2) important ligand-receptor interaction, and 3) exosomal transfer, for cellular communications and morphogenetic or informative gradients.       

Author Response

Comments for Reviewer 2.

Thank you very much for your comments and favorable review. We are glad that you found the review interesting and important for furthering our understanding of the mechanisms of development through cellular interactions, including those that are mediated by morphogenetic and possibly, information gradients.

In accordance with your suggestions we have added several sentences in section 3.0 that address the three issues of, 1) secretion factors (by mentioning the importance of both endocrine and paracrine factors), 2) ligand–receptor interactions that facilitate and make possible cell signaling, and 3) exosomal transfer, which we understood to mean intercellular interactions and more specifically those that cross the extracellular matrix. While the detail of each of these mechanisms would each take a review paper in themselves, we hope that the added sentences addressed your comments. We hope that this has addressed the required changes to the English to which your referred and have also thoroughly proofed the revised manuscript.

We hope that you find the revised manuscript acceptable for publication in Entropy as part of the special issue “Memory and Criticality”.